# Ultrasmall single-layered NbSe$_2$ nanotubes flattened within a chemical-driven self-pressurized carbon nanotube

Yaxin Jiang[1], Hao Xiong [1], Tianping Ying [2], Guo Tian[1], Xiao Chen [1,3] ✉ & Fei Wei [1,3] ✉

Pressure can alter interatomic distances and its electrostatic interactions, exerting a profound modifying effect on electron orbitals and bonding patterns. Conventional pressure engineering relies on compressions from external sources, which raises significant challenge in precisely applying pressure on individual molecules and also consume substantial mechanical energy. Here we report ultrasmall single-layered NbSe$_2$ flat tubes (< 2.31 nm) created by self-pressurization during the deselenization of NbSe$_3$ within carbon nanotubes (CNTs). As the internal force (4–17 GPa) is three orders of magnitude larger than the shear strength between CNTs, the flat tube is locked to prevent slippage. Electrical transport measurements indicate that the large pressure within CNTs induces enhanced intermolecular electron correlations. The strictly one-dimensional NbSe$_2$ flat tubes harboring the Luttinger liquid (LL) state, showing a higher tunneling exponent $\alpha_{NbSe_2@CNT} \approx 0.32$ than pure CNTs ($\alpha_{CNT} \approx 0.22$). This work suggests a novel chemical approach to self-pressurization for generating new material configurations and modulating electron interactions.

Low-dimensional materials have aroused intensive experimental and theoretical interest because of their peculiar electrical, optical, and mechanical properties, which are drastically different from those of their bulk. Subject to dimensional constraints, electrons in 2D materials are trapped within a plane, resulting in a modified band structure that is correlated with the number of layers[1]. Further confinement of electrons within one dimension (1D) leads to the emergence of a collective fermionic state, which is known as the "Luttinger liquid"[2]. It is therefore expected that fabricating low-dimensional materials into other dimensions or changing their configurations allows for further modulation of electronic interactions. Novel physical properties are predicted to arise concomitantly, such as bandgap tuning, metal-insulator conversion, metal-semiconductor conversion, and enhanced thermoelectric properties[3–6].

In addition to the dimensional confinement effect, pressure engineering will be another powerful tool to modify the electronic, magnetic, vibrational, and other intrinsic properties of materials, since the evolution of structures (including non-bonding interactions) is sensitive to external pressure[7]. By squeezing atoms or even forming new bonds, high pressure has always induced exciting physics such as high-temperature superconductivity, super hardness, nonlinear optical character, and insulating electride phases[8–12]. Particularly, pressure applied to an anisotropic crystal structure, such as transition-metal dichalcogenide (TMDs), has aroused intensive experimental and theoretical interest for their potential to induce exotic electronic and topological transitions[13]. Currently, two types of equipment are used to generate high pressure: large-volume presses and diamond-anvil cells that apply static compression, and shock-wave facilities that

[1]Beijing Key Laboratory of Green Chemical Reaction Engineering and Technology, Department of Chemical Engineering, Tsinghua University, 100084 Beijing, China. [2]Beijing National Laboratory for Condensed Matter Physics, Institute of Physics, Chinese Academy of Sciences, 100190 Beijing, China. [3]Ordos Laboratory, 017000 Ordos, Inner Mongolia, China. ✉e-mail: chenx123@mail.tsinghua.edu.cn; wf-dce@mail.tsinghua.edu.cn

generate dynamic compression[14-16], both of which require high energy consumption to apply stress from the outside.

As a typical representative of the 1D system, carbon nanotubes (CNTs) exhibit anomalously strong electron-electron interaction effects, serving as an ideal platform for studying strongly correlated physics. Their natural hollow structure allows the imposing of radial geometric constraints, lowering the dimensions of the materials filling the interior[17-20]. Being built of carbon-carbon *sp2* bonds, one of the strongest covalent bonds, CNTs exhibit superb mechanical strength with Young's modulus of ~1 TPa[21] and thermal stability under 2000 K[22]. Coupled with the chemical inertness of their internal concave surfaces, the ultra-strong carbon bonds allow the CNTs to serve as pressure-holding containers for some aggressive chemical processes. Self-contraction of the CNTs will pressurize the chemical reaction products within the confined channels, changing their energetic stability, generating materials with novel configurations, and modulating the electronic interactions.

Top-down nanoencapsulation within CNTs endows 2D materials with multiple configurations, such as nanoribbons and circular tubes (diameter larger than 3 nm)[18-20]. Because of large bending rigidity resulting from a triple atomic layer, encapsulated TMDs via a chemical vapor transport (CVT) approach are generally single- to few-layered nanoribbons. To date, the reported tubular structures composed of NbSe$_2$ that possess attractive electron-correlated properties are multi-layered, with diameters larger than 30 nm[23]. This study presents the experimental fabrication of ultrasmall single-layered NbSe$_2$ flat nanotubes in the CNTs and reveals that the modulation of electron interactions is linked to the reaction-induced built-up pressure. The encapsulated NbSe$_3$ chains act as reactants under experimental conditions, undergoing spontaneous deselenization followed by atomic rearrangement, with an interior pressure of 4-17 GPa converted by chemical energy. Subjected to strong mutual compression, the CNT and NbSe$_2$ molecules exhibit simultaneous radial deformation, accompanied by a significantly shortened van der Waals (vdW) distance. The resulting NbSe$_2$@CNT exhibits LL behavior with a higher tunneling exponent, demonstrating enhanced intermolecular electron interactions.

## Single-layered flattened NbSe$_2$ nanotube formed within CNT

The encapsulated NbSe$_3$ chains within the CNTs prepared by the CVT method were used as starting materials[17], and the filling ratio of NbSe$_3$ chains is 85-90% (Supplementary Figs. 1-4). After annealing at 873-973 K under 10 vol.% H$_2$/Ar atmosphere, NbSe$_2$ species were obtained by the deselenization reaction, which was confirmed by scanning transmission electron microscopy (STEM), elemental mapping of electron energy loss spectroscopy (EELS), and quantitative chemical analysis of X-ray photoelectron spectroscopy (XPS) (Fig. 1; Supplementary Fig. 5 and Supplementary Table 1). The strict radial geometric constraints of the CNT, coupled with the rolling stress resulting from the shortened Nb-Nb distance during the deselenization process[23], facilitate the formation of the NbSe$_2$ tubular structure. Considering that the system is at atmospheric pressure and moderate constant temperature, the calculated Gibbs free energy difference based on the bulk material is negative ($\triangle G < 0$) (Supplementary Figs. 6-8; Supplementary Tables 2 and 3), and thus the deselenization reaction, NbSe$_3 \rightleftharpoons$ NbSe$_2$ + Se, is thermodynamically favorable that drives it to proceed spontaneously. The high affinity of the nanotube interior for reactant molecules leads to an increased local concentration of reactants, which accelerates the reaction forward[24].

The new NbSe$_2$ phase is caused by the movement and rearrangement of Nb and Se atoms within the confined channels. Upon elevating the temperature, the enhanced diffusion facilitates the free migration of atoms inside the chemically inert CNTs and enables the in-plane growth of NbSe$_2$. According to the statistical analysis of the experimental images before and after deselenization, the number of Nb atoms increases significantly (50-120%) in CNTs with similar equivalent diameters (Supplementary Fig. 9), thus bringing about a significant volume expansion in the radial direction. The intrinsically anisotropic growth behavior of 2D materials will result in expansion along only one direction, thus exerting asymmetric pressure on the CNT inner walls. In turn, the CNT capsule, composed of ultra-strong C-C bonds, spontaneously contracts to counteract the internal expansion, and thus pressurizes its contents (Supplementary Fig. 10). Therefore, the chemical energy is converted into mechanical work

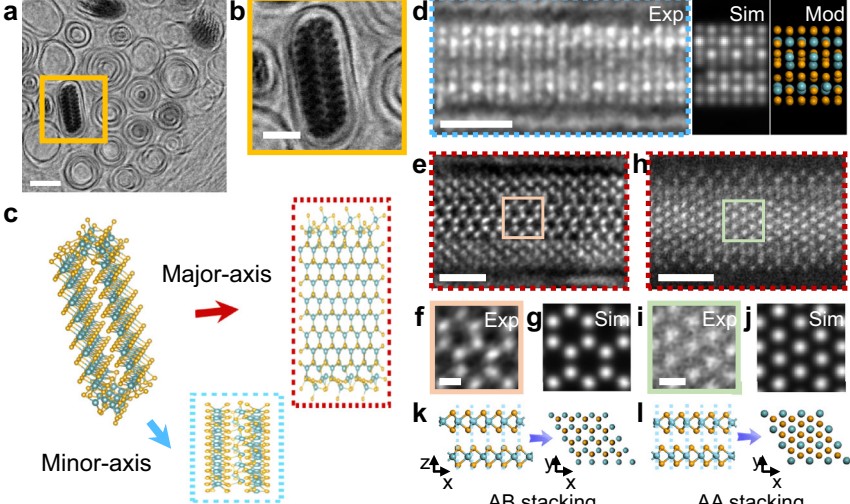

**Fig. 1 | Atomic structure of the single-layered flattened NbSe$_2$ nanotube encapsulated within CNT. a** Atomic-resolved cross-sectional BF images of single-layered flattened NbSe$_2$ nanotube. **b** Enlarged image of the region framed in yellow in **a**. **c** the corresponding optimized atomic models, where the gold and blue atoms represent Selenium and Niobium, respectively. Therein, the red and blue arrows indicate the corresponding images obtained by projection along the major- and minor-axis directions, respectively. **d** Atomic-scale annular dark field (ADF) images with their corresponding simulation and model of NbSe$_2$ flat tube viewed along the minor-axis direction. **e–g** ADF images (**e**), high-magnification images (**f**), and image simulations (**g**) of the AB-stacked NbSe$_2$ flat tube, which shows a 2H phase in the major-axis direction. **h–j** ADF images and corresponding simulation of AA-stacked NbSe$_2$ flat tube with filled honeycomb structure, indicate its 1T phase along the major-axis direction. **k**, **l** Schematic representation of the AB (**k**) and AA (**i**) stacking modes that generate 2H and 1T phase, respectively. The *x*, *y*, and *z* axis correspond to the fragment of the NbSe$_2$ sheet used to model the flat nanotube. Scale bars: 2 nm for **a**, 1 nm for **b**, **d**, **e**, **h**, 200 pm for (**f**, **i**).

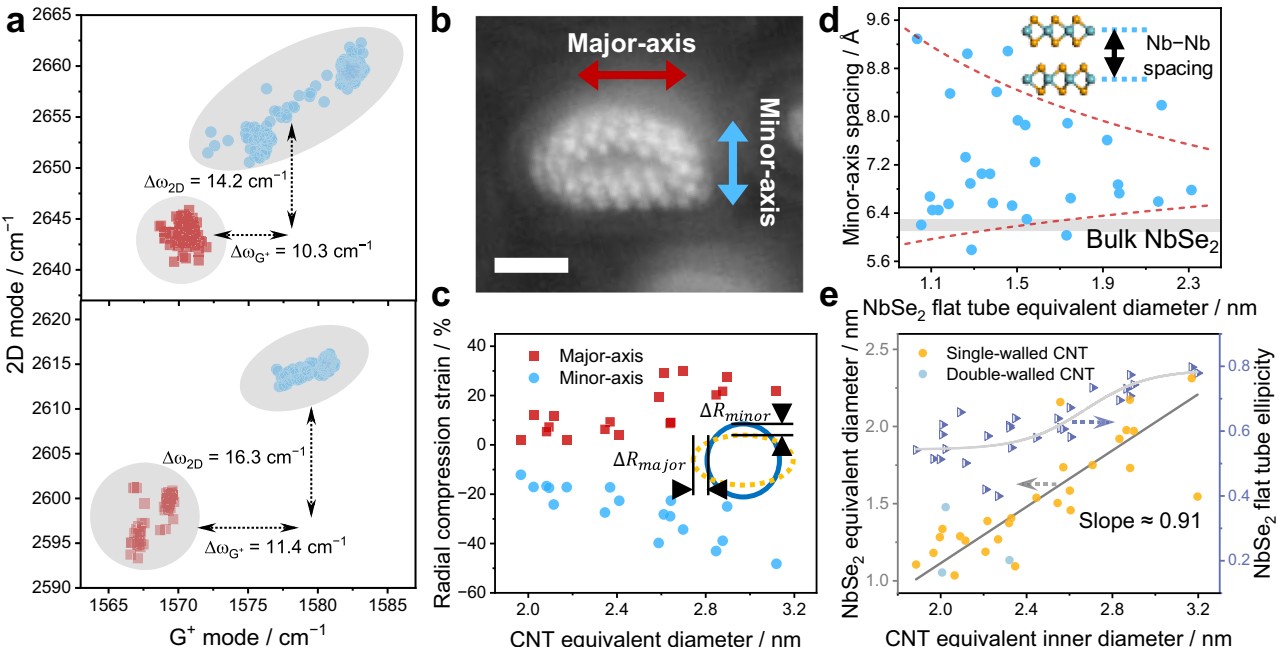

**Fig. 2 | Pressure evaluation of CNT and the resulting lattice deformation of the encapsulated NbSe₂ flat tube. a** Scatter plots of Raman modes and the correlation of 2D- band position plotted against G-band position for samples taken at 532 nm (up) and 633 nm (down) laser excitation wavelength. Pristine CNTs and NbSe₂@CNT flat hetero-tube were marked as red squares and blue circles, respectively. The overlaid gray ellipse demonstrates the approximate distribution region of the scattered points. **b** Atomic-resolved cross-sectional ADF images of single-layered flattened NbSe₂ nanotube. **c** Radial compression strain as a function of CNT equivalent inner diameter in major- (red) and minor-axis (blue) directions. Inset, illustration of the radial displacement of a circular CNT under compression. **d** Relationship between the Nb-Nb spacing of the minor-axis direction (blue points) and the equivalent diameter of the NbSe₂ flat tube, measured from the cross-sectional experimental STEM images. The red dashed lines mark the trend of the Nb-Nb spacing with increasing NbSe₂ diameter, while the gray shading corresponds to the vdW spacing of the bulk counterpart. **e** Linear function of the equivalent diameter (yellow) and S-curve function of the ellipticity (purple) of the monolayer NbSe₂ flat tube versus the equivalent inner diameter of the CNT. The S-curve was obtained by fitting the DoseResp function ($y = A_1 + \frac{A_2 - A_1}{1 + 10^{(LOG_{x0} - x)p}}$) in the class of Sigmoidal functions. Scale bars: 1 nm for **b**.

under synergistic thermodynamic and kinetic effects, favoring the synthesis of species that rarely emerge under ambient conditions. Once the flat tube is formed, the expanding nanostructure will squeeze the outer CNT capsule, behaving like a wine bottle cork. Furthermore, we found that the conversion of the NbSe₃ chains seems to be correlated with the intrinsic chirality of the CNTs, only 5–10% can be converted to the NbSe₂ flat tube. Because of the cork-like morphology, the NbSe₂ flat tube remains stable despite exposure to atmospheric pressure or with the CNT end open.

One critical feature of the obtained NbSe₂ species is that they have radially deformed nanotubes with elliptical shapes (Fig. 1a, b), where Se-Nb-Se sandwiched repeating units are assembled clockwise to form a flat tube, distinguishing this study from all reported coaxial tubes[20,25]. Based on the atomic-resolution Bright-field (BF) STEM image, the representative atomic model of the NbSe₂ flat tube was constructed. At the same time, the two projected directions observed in the STEM experiments of this work were drawn in Fig. 1c, noted as the major- and minor-axis directions. From the minor-axis direction, extra atoms between the upper and lower walls were observed in Fig. 1d, which cannot be classified into common 2D unit cells. It appears as a seamless tube, rather than a double-layer with significant interlayer spacing (Supplementary Figs. 8 and 11). Meanwhile, two typical phases (2H and 1T) can be observed from the major-axis direction, similar to those of their bulk counterparts (Fig. 1e, h). In comparison, the 2H phase presents an empty honeycomb structure (Fig. 1f, g), while the 1T phase presents a filled honeycomb structure (Fig. 1i, j). This discrepancy is accounted for by the different stacking modes of the upper and lower walls of the NbSe₂ flat tube (Fig. 1k, l). A closer look at the middle and edge of the flat tube in Fig. 1e, h, a significant difference in the atomic contrast was observed, with the middle atoms brighter than the edge

ones. Furthermore, the stacking patterns at the edge are also different from those in the middle. These phenomena imply a distinctive curvature at the edges, which further confirms the flattened tubular structure of confined NbSe₂ species.

## Structural correlation of CNT and NbSe₂ flat tubes under built-up pressure

Spontaneous reactions occur in CNTs' confined channels, where chemical energy is converted into mechanical work, producing large pressure from the inside. Previous irradiation studies of CNTs similarly prefigured that high pressure (~40 GPa) could be generated inside the nanotube cavity, while plastically deforming the encapsulated solid materials[26]. Compared to pristine CNTs, the Raman characteristic peak of the NbSe₂@CNT appeared to be upshifted. This indicates the strengthening of the C-C bonds due to a shortened interatomic distance, demonstrating that the CNT is compressed[27]. When multi-region spectra were acquired under laser excitation at 532 and 633 nm, NbSe₂@CNT showed an average G⁺-band upshift of 10.3 and 11.4 cm⁻¹, as well as an average 2D-band upshifts of 14.2 and 16.3 cm⁻¹, respectively (Fig. 2a). These shifts, decoupled by using fractional variations[28], are assumed to be dominated by strain rather than doping effects (Supplementary Fig. 12). Based on the stress-sensitive Raman shifts, the quantified average local strain ranges from ~0.4% to 1.2%[29–31]. Combined with the average modulus of 1 TPa for an individual CNT[21], the local stress inside the CNT cavity reaches 4–12 GPa. Despite the high local stresses, the CNTs were almost free from induced atomic defects, as evidenced by the negligible D-band intensity (Supplementary Fig. 13). Thereby, CNTs that maintain structural integrity can stabilize the internal NbSe₂ flat tubes against structural degradation while exhibiting severe radial plastic deformation under internal large pressure.

From a typical cross-sectional ADF image, both the Se-Nb-Se sandwiched repeating units and the CNT host can be clearly identified (Fig. 2b). With reference to Fig. 2c inset, $\triangle R/R$ is defined as the radial compression strain of the CNT, where $\triangle R$ is the radial compression displacement when the radial pressure is applied. According to the STEM images, the CNT radial strain grows with increasing diameter, with relatively greater compressive strain in its minor-axis direction. Taking the elastic modulus of 36.5 GPa for graphite along the direction perpendicular to the basal plane[32], the strain in the major-axis direction ranges from 2.11% to 30.15%, corresponding to a tensile force of 0.77–11.0 GPa. While the strain in the minor-axis direction ranges from −12.13% to −48.23%, indicating compressive stress of 4.43–17.6 GPa (Fig. 2c). The stresses differ slightly between the nanoscale (individual CNT) and the micron scale (Raman spectra), which can be attributed to the fact that the samples are deformed to varying degrees and some of them are partially circular hollow CNTs (Fig. 1a). Notably, the stress is three orders of magnitude greater than the calculated shear strength between nanotubes (4 ± 1 MPa) and the experimentally found value of one CNT sliding on another (about 4 MPa)[33,34]. As a result, the flat tube inside will be firmly plugged and will not slide out even if both ends of the CNT are exposed to the air.

In an individual CNT, the pressure increases from 4 GPa to 17 GPa as its inner diameter increases, which leads to an aggravated compression of the encapsulated $NbSe_2$ tube, manifested as a change in the Nb-Nb spacing (Fig. 2d). The minor-axis spacing of the $NbSe_2$ tube is consistently larger than the interlayer vdW spacing of bulk counterpart (6.2 ± 0.1 Å) but gradually approaches, which distinguishes it from the layered structure[35]. Besides, it is also implied that the strong interlayer interactions intrinsic to 2D material drive the $NbSe_2$ tubes toward being flattened. Driven by chemical energy, the high curvature edge of the $NbSe_2$ flat tube exerts mechanical work on the CNT inner wall, yielding strong structural correlations as well as strong interactions. All walls of the coaxial nanotubes flatten with strong synchrony, as evidenced by a linear diameter relationship with a slope of 0.91 (Fig. 2e).

Naturally, the magnitude of the pressure exerted by the CNT in turn affects the flattening of the $NbSe_2$ tubes. Therefore, the ellipticity of the $NbSe_2$ nanotube cross-section was introduced for further evaluation, defined as $f = 1 - b/a$, where $a$ and $b$ are the lengths of the major- and minor-axis directions, respectively. The $NbSe_2$ nanotube gradually evolves from nearly circular ($f \approx 0.5$) to elliptical ($f \approx 0.75$) (Supplementary Fig. 14), and its ellipticity appears to grow towards an S-shaped curve (Fig. 2e). By counting the $NbSe_3$ chain number within CNTs of different diameters, the amount of $NbSe_3$ reactants increases nonlinearly with CNT diameter (Supplementary Fig. 4). Smaller diameter (inner diameters <2.2 nm) CNTs accommodate fewer $NbSe_3$ reactants, with less total chemical energy and correspondingly smaller generated internal pressure, resulting in a slow increase of the ellipticity. While larger diameter CNTs have a rapidly increasing reactant density, providing stronger compression of the resulting $NbSe_2$ nanotubes. However, when the equivalent CNT inner diameter exceeds 3 nm, the Nb-Nb spacing of the encapsulated flat tube approaches the vdW spacing and cannot be reduced further (Fig. 2d), and the increase in ellipticity is moderated again. Moreover, under severe radial geometrical constraints, the experimentally obtained $NbSe_2$ nanotube shows diameters not exceeding 2.31 nm, which is lower than all reported theoretical or experimental values for TMD circular tubes.

## Energy-favored isolation of strained single-layered $NbSe_2$ flat tube

To confirm the energetic stability of the flat tube, we performed first-principle calculations based on DFT regarding the various possible $NbSe_2$ geometric configurations. Four initial $NbSe_2$ configurations with different cross-sectional units ($n$, $n = 6$–24) isolated in a vacuum were simulated, including single-layer, double-layer, circular tube, and flat tube (Supplementary Figs. 15–18). Atomic positions of these configurations are derived from the bulk solid and then fully relaxed by minimizing the total energy. Moreover, the pressure shows a negligible effect on the formation energy of the fully relaxed configurations (Supplementary Figs. 19–22). For the different $NbSe_2$ layered configurations, a $1/n$ dependence of the formation energy on the number of cross-section units was observed (Fig. 3a). This suggests a negligible dependence of their dangling bond energy on the layers' width, with an identical geometry at the edges. While for different $NbSe_2$ tubular configurations, the formation energy roughly follows $-1/n^2$ rules. It indicates an increase in stability with increasing $n$, which reduces their curvature to lower the strain energy, as known for CNTs[36]. After $n$ increases to a certain value ($n > 10$), the formation energy is much lower for the tubular structure. This difference can be simply explained by the competition between the dangling bond energy of layer edges and the bending energy of the tubular structure.

When $n = 9$–24, the $NbSe_2$ flat tube presents the most thermodynamically stable conformation, which is attributed to the saturation of the dangling bonds and the vdW adhesion in the flat zone. Beyond a certain value of $n$, the vdW interactions are offset by the local stress due to the increased ellipticity. Therefore, an intersecting point is expected, leading to their stabilization into circular tubes. As shown in the inset of Fig. 3b, the diameter of a $NbSe_2$ circular tube increases linearly with $n$. Equivalently, the formation energy of the circular tube obeys the rule of $-1/D^2$ that predicts the layer bending within the framework of classical elasticity theory. Assuming that flat and circular tubes of the same $n$ possess the same equivalent diameter, the formation energy of flat tubes satisfies a similar $1/D^2$ relationship. Further, the corresponding formation energy regions that are more favorable for flat tubes can be derived (purple shading), with the critical diameters identified as 1.02 and 2.45 nm. Whereas the experimentally observed equivalent diameters spanned from 1.03 to 2.31 nm (Fig. 2e), falling exactly within the calculated diameter range. Therefore, it can be expected that well-defined $NbSe_2$ configurations will be acquired by strictly regulating the inner diameter of CNTs. According to the relationship between the diameters of CNT and $NbSe_2$ flat tubes fitted in Fig. 2e, the CNT diameters in the experiments should be strictly regulated at 1.90–3.46 nm for synthesizing single-layered $NbSe_2$ flat tubes rather than layers or circular tubes.

When bending into a tubular structure, the symmetry of the initial unit is broken, manifesting as a modification of the Nb-Se bond length. On average, the outer Nb-Se bonds of the "Se-Nb-Se" sandwich structure of the $NbSe_2$ circular tube were elongated, while the inner bonds were shortened (Fig. 3c), resulting in significant formation energy. With a gradual increase of $n$, the bond lengths will simultaneously approach 2.626 Å of the bulk phase[37], yielding a more stable structure with small strains. Unlike the relatively uniform change of the Nb-Se bond length in the circular tube (Supplementary Fig. 23), a $NbSe_2$ flat tube undergoes negligible bond compression (<3%) in the major-axis direction, with the bond length varying mainly at the edges of the minor-axis (Fig. 3d). Notably, the maximum compression and elongation of the Nb-Se bonds at the edges reaches even −9.01% and 10.18%, respectively, at $n = 20$. Furthermore, the bond length changes can be used to estimate the local pressure. Based on the measured compression modulus of 114 GPa for the bulk $NbSe_2$[38], about 3–10% strain in the atomic model originates from a local pressure of 3.42–11.4 GPa. This value is highly comparable to the stresses measured from Raman spectra (4–12 GPa) and cross-sectional STEM images (4.43–17.6 GPa).

## Enhanced electronic correlations in the 1D system under large pressure

Pressure effects modify not only the bonding interactions, but also the non-bonding interactions (vdW forces). The vdW interaction can be defined by the conventional 6–12 type Lennard Jones (LJ) potential[39],

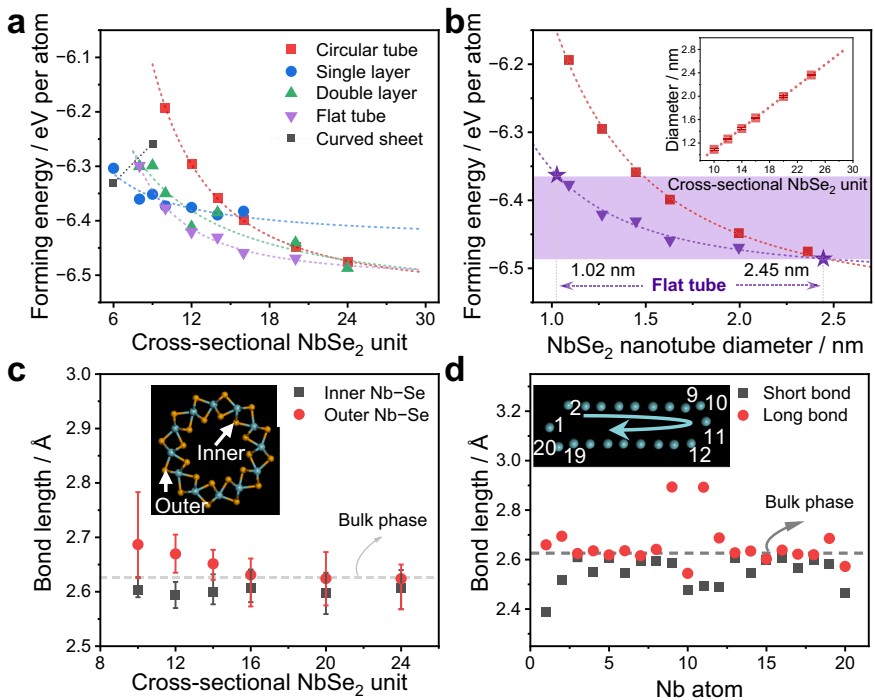

**Fig. 3 | Energetically stabilization of flattened NbSe₂ nanotube. a** Calculated forming energy of varied NbSe₂ configurations (energy per atom) as a function of the number of NbSe₂ units in their cross-section. The formation of single- or double-layer configurations roughly satisfies the $1/n$ dependence, while flat and circular tubes obey the $1/n^2$ rule. **b** Forming energy of NbSe₂ tubular structures as a function of their equivalent diameter, in which circular and flat tubes are in red and purple, respectively. The purple-shaded region represents where their energy favors the isolation of flat tubes, as derived from **a**. The inset shows that NbSe₂ diameters for different circular tubes depend on their cross-sectional unit number

measured by the relaxed atomic model, following a linear relationship. **c** Average Nb-Se bond lengths of inner and outer layers of "Se-Nb-Se" sandwich structures of NbSe₂ circular tubes at different $n$. The upper limit of the error bars is the maximum of all Nb-Se bond lengths in the optimized NbSe₂ circular tube model, the lower limit is the minimum bond length, and the red dot is the average bond length. The inset shows a typical circular tube model with arrows indicating the inner and outer Nb-Se bonds. **d** Bond length distribution of short and long Nb-Se bonds in the flat tube along the clockwise direction of Nb atoms shown in the inset at $n = 20$.

which is expressed as:

$$E_{vdW} = D_{IJ}\left[-2\left(\frac{x_{IJ}}{x}\right)^6 + \left(\frac{x_{IJ}}{x}\right)^{12}\right] \qquad (1)$$

Where $D_{IJ}$, $x_{IJ}$, and $x$ signify the energy parameter, vdW distance parameter and interatomic distance, respectively. Theoretical studies have determined that the well depth ($D_{IJ}$) of Se and C atoms is 7.58 meV and the vdW Se-C bond length ($x_{IJ}$) is 4.024 Å. According to the STEM experimental images, the spacing between the CNT and the outermost Se atom was 2.815–3.520 Å (Fig. 4a, b), which is 12.5–30.0% smaller than the theoretical value[39]. Consequently, the vdW interaction between CNT and NbSe₂ flat tubes grows exponentially with decreasing distance at high internal pressure, which is expected to prevent internal structural sliding and deliver strong intermolecular electronic correlations.

Temperature-dependent resistivity (R–T) measurements were performed with a standard four-probe method to investigate the electron transport behavior and the electronic correlation of heterogeneous samples. Nonlinear R–T relationships were observed in different fibrous samples (Fig. 4c). Notably, the NbSe₃@CNT exhibits a distinct curve with both metallic and semiconducting characteristics. The presence of internal metallic NbSe₃ chains led to an appreciable increase in the density of states near the Fermi energy compared to pristine CNTs, resulting in a semiconducting to metallic transition. Upon initial observation, the R–T curve for the transformed NbSe₂@CNT appears to be similar to that of unfilled CNTs. However, its divergence behavior at lower temperatures is quite different.

The electrical behavior of CNTs is influenced by their chirality index $(n, m)$, and when MOD[$(n-m)$, 3] ≠ 0, the system exhibits Luttinger-liquid (LL) behavior, in which the energy states of electrons near the Fermi level ($E_F$) are strongly perturbed by Coulomb interactions[2]. The power-law plot of unfilled CNTs shows the difference in conductance of different samples, suggesting dissimilarities in the number and distribution of CNT-CNT junctions in the ensembled samples (Fig. 4d). However, a consistent Luttinger exponent $\alpha_{CNT}$ of 0.64 was demonstrated at high temperature (10–100 K), which is in agreement with previous research[40], implying a strong response to the intrinsic electron transport properties of CNTs. Remarkably, the electrical curve of NbSe₂@CNT appears to be steeper approaching the low-temperature limit. At low temperatures (2–10 K), the fitting parameter of NbSe₂@CNT gives a much larger exponent of $\alpha_{NbSe_2@CNT} \approx 0.32$ than that of unfilled CNTs ($\alpha_{CNT} \approx 0.22$) and NbSe₃@CNT ($\alpha_{NbSe_3@CNT} \approx 0.18$) (Fig. 4e; Supplementary Figs. 24 and 25). Despite the filling rate of only 5–10%, the average electrical behavior of NbSe₂@CNT demonstrates that merely a minimal amount of NbSe₂ also greatly enhances the electron correlation of the current system. While the typical 2D bulk NbSe₂ exhibits distinctly different metallic behavior, with a $T^{-3}$ dependence of its conductance at $T < 25$ K[41]. Additionally, the exponential relationship between conductance and temperature extends to a higher temperature range of 2–100 K for NbSe₂@CNT (Supplementary Fig. 24). The Luttinger parameter $K$ of NbSe₂@CNT, a single dimensionless parameter reflecting the electron-electron interaction strength, even reaches 0.23, which is less than the limit of 0.25 for the very strong interaction (see Supplementary Information)[42]. These findings provide a viable strategy to modulate the strength of electron correlation

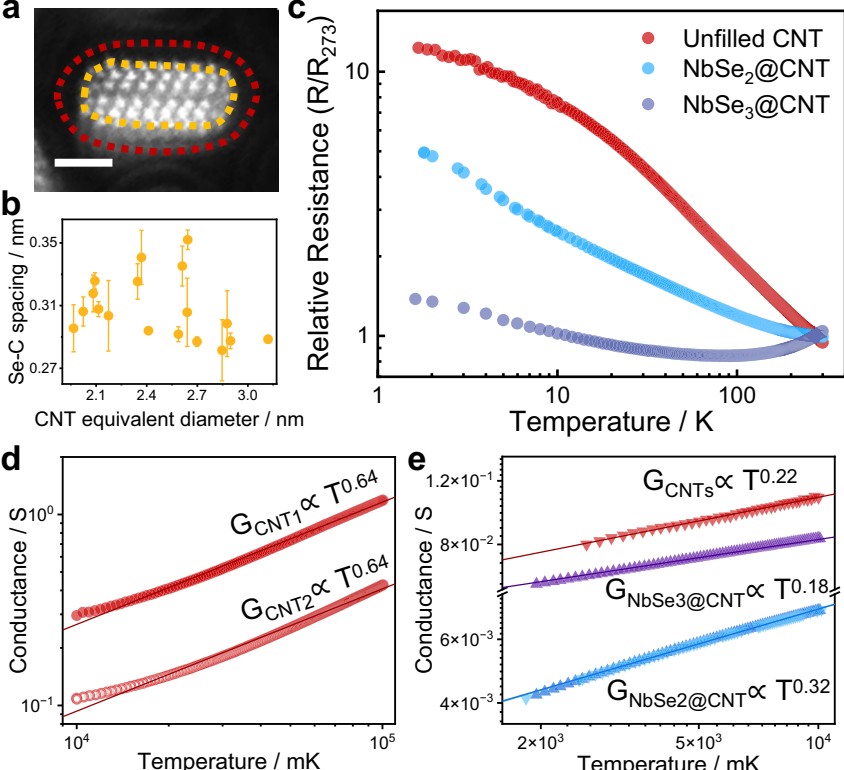

**Fig. 4 | Electronic properties of the flattened NbSe₂@CNT hetero-tube.**
**a** Representative cross-sectional ADF image of NbSe₂ flat nanotube, where the outermost Se atoms are depicted with orange ellipses and the CNT is traced with red ellipses. Scale bars, 1 nm. **b** Average Se-C distance which was counted in CNTs of different inner diameters. The error bar arises from the slight difference in spacing along the major- and minor-axis directions. **c** Temperature-dependent resistance of unfilled CNT (red), NbSe₂@CNT (blue), and NbSe₃@CNT (purple) fibers. **d** Power-law relation of conductance G with temperature ($G\sim T^{\alpha}$) for two pure CNT samples from 10–100 K, obtaining a fitting exponent $\alpha = 0.64$ from the red fitting curve. Inset, an optical image of CNT samples with the silver electrode deposited on the top of the samples. **e** Power-law relation for CNTs, NbSe₃@CNT, and NbSe₂@CNT from 2–10 K, giving a fitting exponent of 0.22, 0.18, and 0.32, respectively.

in a confined 1D system and open up new perspectives on self-pressurization from the interior.

## Summary

We demonstrate a feasible approach for applying high pressures to a single nano molecule through internal chemical reactions within CNTs. As a result, a thermodynamically stable novel morphology of TMDs, single-layered flat tube, emerges within a synchronous radially deformed CNT. The experimentally found interaction forces (4–17 GPa) are three orders of magnitude stronger than the intertube shear strength of CNTs, locking the internal NbSe₂ flat tube and thereby maintaining the CNTs pressurized. Both bonding and non-bonding (vdW) interactions are altered upon strong compression, which in turn modifies the competition between the kinetic energy of the electrons and various electrostatic Coulomb interactions. The obtained NbSe₂@CNT resembles a Luttinger-liquid, showing very strong electron interactions compared to the pristine CNTs. This approach of self-pressurization by chemical reactions within the confined channel opens new horizons for imposing large pressures on materials, which leads to the emergence of novel nanostructures with enhanced electron interactions, potentially exhibiting fascinating physical properties.

## Methods

### Synthesis of single-layered NbSe₂ flat nanotubes encapsulated within CNTs

Carbon nanotubes (CNTs) were grown by floating catalyst chemical vapor deposition (FCCVD) method with methane as the carbon source. These as-grown CNTs are mostly with a concentrated inner diameter distribution within 2–3 nm. Then, the CNTs were annealed in air at 510 °C for 15 min to open the end caps. The chains of NbSe₃ were synthesized within CNTs to serve as the starting material[13]. Briefly, stoichiometric quantities of Nb and Se powders (-20 mg in total) together with 1–2 mg of as-prepared CNTs were sealed under vacuum in a quartz ampoule. The ampoule was heated and kept at 690 °C for several days, and then gradually cooled to room temperature over 3–9 days. To transform the NbSe₃ chains inside CNTs, the samples were treated at 700 °C for 1–20 min in 100 mL/min of 10% H₂/Ar. Control heating procedures were performed to ensure minimal damage to the CNT sidewalls from H₂.

### Scanning transmission electron microscopy (STEM) sample preparation and characterization

As-prepared samples were sonicated in ethanol for 30 min to diminish bundle aggregation, followed by dropcasting the dispersion onto a copper TEM microgrid. Because of their high aspect ratio, the samples lie on the microgrid surface, which is then viewed from the side-view projected direction. Whereas for cross-sectional observation, the samples were prepared by means of focused ion beam (FIB) milling. At an accelerating voltage of 30 kV along with a gradual decrease in current from the maximum of 2.5 mA, the cross-sectional lamellae were thinned to 60 nm thick. Afterward, fine polishing was performed with a small current of 40 mA at an accelerating voltage of 2 kV.

A DCOR+ spherical aberration-corrected STEM (FEI Titan Cubed Themis G2 300) was used to collect the BF and ADF images. The instrument was operated at 300 kV with a convergence semi-angle of 15 mrad and the collection angles were set as 5 and 17–102 mrad to acquire BF and ADF images, respectively. This STEM was aligned with

proper aberration coefficients using a standard gold sample, and the aberration coefficients are C1 = 3.78 nm; A1 = 6.04 nm; A2 = 44.2 nm; B2 = 45.6 nm;  C3 = 429 nm;  A3 = 213 nm;  S3 = 347 nm;  A4 = 12 μm, D4 = 20.6 μm, B4 = 15.9 μm, C5 = −940 μm, A5 = 252 μm. The obtained images are applied with a Gaussian filter to denoise.

The STEM simulations were conducted via the abTEM open-source software based on the multislice algorithm[43,44]. A slice thickness was set to 0.5 Angstrom to improve the accuracy. The convergence angle (15 mrad), collection angle (17–102 mrad), and probe step size (0.25 Angstroms) were used as the same in the ADF experiments.

## Pseudopotential density functional theory (DFT) calculations

The atomic models of the $NbSe_2$ single-layer or double-layer were built based on bulk $NbSe_2$ atomic structure (with lattice parameters $a = 0.3449$, $c = 12.550$ nm). The principle of constructing an atomic model of $NbSe_2$ circular nanotubes coincides with that of graphene-derived CNTs, in which $NbSe_2$ monolayers are mapped onto the cylindrical surface to form achiral armchair "triple-walled" Se-Nb-Se tubes. Based on the optimized circular tube models, $NbSe_2$ flat tube atomic models were constructed by modifying the atomic positions in accordance with the experimental images. Because of the axial periodic boundary conditions, these initial models add a distance of about 6 Å in the "non-periodic" (radial) direction to eliminate unphysical interactions.

Structural optimizations of all systems were performed with density functional theory using Vienna Ab initio Simulation Package (VASP)[45]. The electron exchange and correlation energy were modeled by using the Perdew–Burke–Ernzerhof (PBE) functional form of generalized gradient approximation (GGA)[46]. The electron-core interaction was described by the projector augmented wave (PAW) method[47], and a kinetic energy cutoff of 450 eV was adopted for the plane waves. For the geometry optimization and energy computations, Gaussian smearing of 0.2 eV was applied to the orbital occupation. All structures were optimized until force components were less than 0.02 eV/Å, and a convergence threshold of $10^{-6}$ eV for the iteration in the self-consistent field (SCF).

## Spectroscopy characterization

Raman spectroscopy was conducted with a Horiba HR 800 Raman spectrometer equipped with lasers operated at wavelengths of 532 and 633 nm. Measurements were taken at room temperature under ambient conditions. The G and 2D-bands of each Raman spectra were fitted with Lorentzian line shape.

XPS was performed using monochromatic Al Kα radiation. For the measurements, Samples with dimensions of 5 mm × 10 mm were mounted on Mo sample holders. The energy resolution is better than 0.45 eV, and the atomic ratio of each element in the sample was also measured.

## Electrical transport measurements

The electrical resistivity was measured on individual fiber samples in a He flow pulse-tube cryocooler between 2 and 300 K. Each fiber sample was prepared to be about 1 mm long and their diameters were typically around 140 μm. The electric contacts were conducted by gluing a gold wire to the sample surface with silver paint, and the distances were uniform to allow homogeneous current distribution. For each sample, the input current was kept at 10 μA to avoid self-heating, and the resistance was measured within its Ohmic resistance range with a resolution of one part in $10^5$.

## Statistics and reproducibility

No statistical method was used to predetermine the sample size. No data were excluded from the analyses. The experiments were not randomized. The investigators were not blinded to allocation during experiments and outcome assessment.

## Reporting summary

Further information on research design is available in the Nature Portfolio Reporting Summary linked to this article.

## Data availability

All data needed to evaluate the conclusions in the paper are present in the main text and/or the Supplementary Information. Source data in this study are provided in the Source Data file. Source data are provided in this paper.

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

## Acknowledgements

This work was supported by the Ministry of Science and Technology of China (2022YFA1203301, F.W., 2021YFA1401800, T.Y.), the National Natural Science Foundation of China (No. 22275110, 22322203, X.C.), and the National Key Research and Development Program of China (No. 22238004, F.W. and X.C.). This research was sponsored by the Tsinghua-Toyota Joint Research Fund (X.C. and F.W.). The authors are grateful to the Tsinghua National Laboratory for Information Science and Technology for assistance with the energy simulation.

## Author contributions

X.C. and F.W. conceived this project and designed the studies; Y.J., H.X., and X.C. performed the electron microscopy experiments and data analysis; Y.J. and G.T. carried out the first-principles calculations; H.X. performed the simulation of iDPC-STEM images; T.Y. performed the electrical transport measurements; All authors are involved in the data analysis; Y.J. wrote the manuscript with the help of others.

## Competing interests

The authors declare no competing interests.
