## [Peer Review File · Nature Communications]

Reviewers' Comments:

Reviewer #1:

Remarks to the Author:

The authors have done an impressive study where they present the experimental fabrication of ultrasmall single-layered NbSe₂ flat nanotubes in the CNTs, and show that the electronic interactions can be modified by reaction-induced pressure engineering. As a result, the authors could obtain thermodynamically stable novel morphologies of TMDs within the CNTs. The authors successfully modified both the bonding and nonbonding interactions as well as the electrostatic interactions. This report is of importance and will be helpful to the community.

Few suggestions:

1. The introduction needs to significantly improved - lot more details regarding the background of the problem and why this work is relevant should be discussed. Atleast some information regarding the Luttinger liquid state should be discussed.
2. I found it difficult to understand what the authors did differently from previous studies. The authors should clearly discuss/emphasize the experimental conditions that were necessary to obtain ultrasmall single layers of NbSe₂ instead of multilayers.
3. Can the authors provide a general protocol to obtain such single layer NbSe₂?
4. Can the authors please explain why the middle atoms were brighter than the edge ones?
5. Is it possible to correlate the degree of pressure with the type of stacking at the edge and in the middle?

The article is publishable subject to reasonable explanations/clarifications to the above.

Reviewer #2:

Remarks to the Author:

Y. Jiang et al. report the formation of flattened ultrasmall NbSe₂ nanotubes inside carbon nanotubes. The authors argue that the NbSe₂ nanotubes are under high pressure due to the carbon nanotube container. Although the atomic resolution STEM imaging and DFT simulations have been reasonably well conducted, the manuscript's main arguments lack of convincing data. In particular, the sample state under the high pressure and interpretation of the electrical transport measurement are not convincing to this reviewer.

1. The authors argue that the flattening of NbSe₂ is originated from the pressure built inside the carbon nanotube. The authors estimated the pressure using the degree of deformation and elastic modulus. However, the flattening of the nanotube itself cannot be a good way to assess the pressure on samples. There have been numerous previous reports on the flattening of carbon nanotubes. (For examples, see Fully collapsed carbon nanotubes, *Nature* 377, 135–138 (1995); Closed-Edged Graphene Nanoribbons from Large-Diameter Collapsed Nanotubes, *ACS Nano* 6, 6023–6032 (2012), etc.) The nanotube can undergo deformation or flattening because of the thermodynamic reasons. The deformation itself could have nothing to do with the pressure.
2. Raman spectroscopy on ensemble samples was presented as another indicator for pressure. However, Raman measurement on ensemble samples, which contains various states of sample encapsulation and different diameters and number of walls in carbon nanotube containers, is not convincing. The Raman peak shift from NbSe₂ encapsulation could also originate from different reasons, such as the doping effect. To remove the uncertainty, the authors must also consider the effect of doping and others. Moreover, the authors must present better Raman data with well-defined sample configurations, such as the same-diameter carbon nanotube with/without NbSe₂ encapsulation.
3. The authors also performed the electrical measurements. Although it seems that the authors tried their best, the electrical transport measurements on the carbon nanotube bundles are not a good way to discuss the intrinsic properties of carbon nanotubes. For example, the nanotube-nanotube junction will significantly influence the electrical transport properties but was not

considered at all in the manuscript. In particular, the surface functionalization or doping from the NbSe₂ encapsulation process may cause unexpected behaviors.

4. Finally, it needs to be clarified the origin of the existence of high pressure from the synthesis process. The authors explained that the deselenization step induces pressurization, but why? Could the starting material, NbSe₃, be also under pressure? What is the deriving mechanism for such an energetically unfavored process (to a high pressurized state) if not?

Reviewer #3:

Remarks to the Author:

In this manuscript, the authors report on the formation of flat NbSe₂ tubes from NbSe₃ chains inside carbon nanotubes upon annealing. The structure of the flat tubes is determined based on high resolution transmission electron microscopy images, which in turn serve as the basis for simulations. The authors further measure the resistance of samples without NbSe and after being filled with the chains and after the formation of the NbSe₂ tubes. The observed lowering of conductance is argued to be a result from enhanced electron correlation.

My only concern with the manuscript is that no statistical analysis is provided to demonstrate the concentration of either the chains or the resulting flat tubes inside the nanotubes (ie, what percentage of the tubes is filled). This however has an important role in trying to understand whether the measured differences in conductance are indeed associated with filling of the tubes (and hence the NbSe structures) or whether it may arise from other sources related to f.ex. high-temperature annealing. Judging from Fig. 1a it appears that the filling rate is not very high, since this image is presumably from an area displaying the observed structures the best.

Only after this point has been clarified, with appropriate evidence and discussion, the manuscript can in my view be considered for publication. Additionally, spectroscopic characterization of the structures (for example through EELS mapping) would provide additional evidence supporting the interpretation of the atomic structure of the NbSe₂ tubes.

Referee #1

Thank you for your recommendation of our manuscript. We are excited to see your very positive comments on the novelty and significance of this work. Your comments and suggestions show your high-degree expertise on nanomaterials preparation and really help us to improve our manuscript efficiently. We carefully studied these comments and made modifications which we hope meet your approval. After this revision, we sincerely ask you to reconsider our revised manuscript for the further publication.

1) The introduction needs to significantly improved - lot more details regarding the background of the problem and why this work is relevant should be discussed. At least some information regarding the Luttinger liquid state should be discussed.

Reply: Thanks very much for your suggestions, which are very helpful to improve the quality of this article. We tried our best to improve the introduction and made some changes to the manuscript. We have enriched the background and further emphasized the importance and relevance of this work, and have added the discussion of Luttinger liquid state that you mentioned. Here are the changes and the revised manuscript marked in blue as follows:

“Low-dimensional materials have aroused intensive experimental and theoretical interest because of their peculiar electrical, optical, and mechanical properties, which are drastically different from those of their bulk. Subject to dimensional constraints, electrons in 2D materials are trapped within a plane, resulting in a modified band structure that is correlated with the number of layers. Further confinement of electrons within one dimension (1D) leads to the emergence of a collective fermionic state, which is known as the "Luttinger liquid". It is therefore expected that fabricating low-dimensional materials into other dimensions or changing their configurations allows for further modulation of electronic interactions. Novel physical properties predicted to arise concomitantly, such as bandgap tuning, metal-insulator conversion, metal-semiconductor conversion, and enhanced thermoelectric properties.

In addition to dimensional confinement effect, pressure engineering will be another powerful tool to modify the electronic, magnetic, vibrational, and other intrinsic properties of materials, since the evolution of structures (including nonbonding interactions) is sensitive to external pressure. By squeezing atoms or even forming new bonds, high pressure has always induced exciting physics such as high-temperature superconductivity, superhardness, nonlinear optical character, and insulating electrified phases. Particularly, pressure applied to an anisotropic crystal structure, such as transition-metal dichalcogenide (TMDs), has aroused intensive experimental and theoretical interest for their potential to induce exotic electronic and topological transitions. Currently, two types of equipment are used to generate high pressure: large-volume presses and diamond-anvil cells that apply static compression, and shock-wave facilities that generate dynamic compression, both of which require high energy consumption to apply stress from the outside.

As a typical representative of the 1D system, carbon nanotubes (CNTs) exhibit anomalously strong electron-electron interaction effects, serving as an ideal platform for studying the strongly correlated physics. Their natural hollow structure allows the imposing of radial geometric constraints, lowering the dimensions of the materials

filling the interior. Being built of carbon-carbon sp² bonds, one of the strongest covalent bonds, CNTs exhibit superb mechanical strength with a Young's modulus of ~1 TPa and thermal stability under 2000 K. Coupled with the chemical inertness of their internal concave surfaces, the ultra-strong carbon bonds allow the CNTs to serve as pressure-holding containers for some aggressive chemical processes. Self-contraction of the CNTs will pressurize the chemical reaction products within the confined channels, changing their energetic stability, generating materials with novel configurations and modulating the electronic interactions.

Top-down nanoencapsulation within CNTs endows 2D materials with multiple configurations, such as nanoribbons and circular tubes (diameter larger than 3 nm). Because of large bending rigidity resulting from a triple atomic layer, encapsulated TMDs via a chemical vapor transport (CVT) approach are generally single- to few-layered nanoribbons. To date, the reported tubular structures composed of NbSe₂ that possess attractive electron-correlated properties are multi-layered, with diameters larger than 30 nm. This study presents..."

2) I found it difficult to understand what the authors did differently from previous studies. The authors should clearly discuss/emphasize the experimental conditions that were necessary to obtain ultrasmall single layers of NbSe₂ instead of multilayers.

Reply: Thank you for your comments, and we feel sorry for our writings. In contrast to the one-step chemical vapor transport method reported in previous studies, our work employs a pre-filled sample as a precursor and subsequently obtains a new phase of NbSe₂ through a chemical reaction inside a carbon nanotube. In this case, the diameter of the carbon nanotube determines the morphology of the encapsulated NbSe₂ species. As demonstrated in Fig. 2e, NbSe₂ flat tubes were found only in CNTs with diameters of 1.9-3.2 nm. Therefore, the diameter of the CNTs is the key to obtain ultrasmall flat tubes of NbSe₂ rather than multilayers. We have added "According to the relationship between the diameters of CNT and NbSe₂ flat tubes fitted in Fig. 2e, the CNT diameters in the experiments should be strictly regulated at 1.90-3.46 nm for synthesizing single-layered NbSe₂ flat tubes rather than layers or circular tubes." in the manuscript to discuss the experimental conditions more clearly, and we hope it would be more understandable.

Fig. 2e Relationship between the equivalent diameter (yellow) and ellipticity (purple) of the single-layer NbSe₂ flat tube versus the equivalent inner diameter of the CNT.

3) Can the authors provide a general protocol to obtain such single layer NbSe₂?

Reply: Thank you for your suggestion. We presented the preparation process in detail in the Supplementary Material, and consider that the following steps are crucial:

1. Preparing CNTs with diameters concentrated at 2-3 nm, along with μm -level lengths and few defects, in order to reduce loss of filling materials from defects or ends during inward contraction and extrusion.
2. Annealing the CNTs in air at 510 °C for 15 minutes to open their end caps, but rarely destroy the sp² carbon bonding network in their sidewalls.
3. Open-ended CNTs were mixed with Nb and Se powders (stoichiometric ratio 1:3), placed in a quartz ampoule under high vacuum, heated at 690°C for several days and then slowly cooled to obtain the starting material NbSe₃@CNT. This step is consistent with that reported by Zettle's group (Pham T, Oh S, STETZ P, et al. Science, 2018, 361, 6399, 263-266).
4. NbSe₃@CNT was further treated in a 10% H₂/Ar atmosphere at a flow rate of 10 mL/min and a temperature of 973 K. It is important that the heating time must be controlled at 1-20 min to reduce the damage of H₂ to the CNT sidewall.

4) Can the authors please explain why the middle atoms were brighter than the edge ones?

Reply: Thank you for your question. The annular dark field (ADF) imaging in a STEM has proved to be efficient for atom-by-atom identification at high spatial resolution. Under the experimental conditions where the angular range of the collected electrons was about 17–102 mrad half-angle, the collected dark-field signal in ADF STEM is caused by Rutherford scattering from the partially screened atomic

nucleus, which increases with the atomic number (Z) as about $Z^{1.34}$ (Fig. R1). Here, the middle atoms in the side-view projection image are stacked Nb/Se atoms, while the edge atoms are individual Se atoms, with the significantly larger Z of the middle atoms leading to their greater brightness.

Figure R1. Plot of the average ADF intensities of the different elemental atoms versus their atomic number, Z .

5) Is it possible to correlate the degree of pressure with the type of stacking at the edge and in the middle?

Reply: Thank you for your suggestion. There are clear stacking types, including AA and AB stacking mode, in the side-view projections STEM images. However, these stacking patterns can only be precisely confirmed from the side view. From cross-sectional images where pressure can be measured, it is difficult to accurately identify stacking patterns, especially at the edges. Moreover, the encapsulated NbSe₂ tube is not necessarily in a fully compressed state (Fig. 2b), and therefore cannot be attributed to a certain stacking type, which further limits our ability to correlate the degree of pressure with the type of stacking. We believe that there must be a relationship between the inner diameter/chirality of the CNTs and the type of stacking of NbSe₂, and further correlating this with the pressure magnitude would indeed be relevant. This requires us to synthesize a variety of CNTs with a single chirality, which is a recognized challenge in the field of CNT synthesis. We will continue our efforts to explore this in the future on the basis of our current work.

Fig. 2b Cross-sectional ADF images of single-layered flattened NbSe₂ nanotube that are not fully compressed.

Referee #2

Thank you for your time and efforts on our manuscript. We are glad to see your positive comments on the significance of our atomic resolution STEM characterization and DFT simulations. In this work, we prepare a novel phase of single-layered, flattened NbSe₂ nanotubes by deselenization reaction inside the CNTs, and reveal the atomic structure of NbSe₂ novel phase by resolving STEM images with different views. The drastic radial deformation of the CNT host, the large shift of the Raman characteristic peaks, the change of the bond length of the Nb-Se bond at the flat tube edge, and the significant shortening of the van der Waals distance between Se and C all together point out that the samples state is under pressure. Based on your suggestions, we have supplied a decoupling of mechanical strain and charge doping in Raman spectroscopy, and revised the discussion on the origin of the existence of pressure from the synthesis process. Please see details in the following replies to your comments point to point. We will appreciate your suggestions on further improving this manuscript.

1) The authors argue that the flattening of NbSe₂ is originated from the pressure built inside the carbon nanotube. The authors estimated the pressure using the degree of deformation and elastic modulus. However, the flattening of the nanotube itself cannot be a good way to assess the pressure on samples. There have been numerous previous reports on the flattening of carbon nanotubes. (For examples, see Fully collapsed carbon nanotubes, *Nature* 377, 135–138 (1995); Closed-Edged Graphene Nanoribbons from Large-Diameter Collapsed Nanotubes, *ACS Nano* 6, 6023–6032 (2012), etc.) The nanotube can undergo deformation or flattening because of the thermodynamic reasons. The deformation itself could have nothing to do with the pressure.

Reply: We appreciate your new insights on this work, and we agree that NbSe₂@CNT is in thermodynamic equilibrium. However, as the action of forces is mutual, the overall thermodynamic equilibrium of the CNTs and NbSe₂ flat tubes does not contradict the pressure that exist between them. And we respectfully disagree with your statement that the flattening of the CNTs has nothing to do with pressure. We believe that the radial deformation of CNTs is the most visual response to pressure, based primarily on the following facts:

1. Only large-diameter CNTs (> 6 nm) can thermodynamically flatten by spontaneous collapse. There have been a number of reports of carbon nanotube flattening. The reported collapsed CNTs are all of large-diameters, with most of the inner diameters larger than 10 nm (Chopra, N., Benedict, L., Crespi, V. et al. *Nature*, 1995, 377, 135-138). Zhang et al. elucidated the strong diameter dependence of the CNTs configurations. Single-walled CNTs are stable cylinders at diameters less than 2.42 nm, equilibrium between cylindrical and collapsed configurations at diameters of 2.42-6.24 nm (whereas cylinders are energetically more favorable), and stable collapsed configurations at diameters greater than 6.24 nm (Zhang S, Khare R, Belytschko T. et al. *Physical Review B*, 2006, 73, 7:075423). For multi-walled carbon nanotubes with wall numbers ≥ 2 , the critical diameter of collapse will be even larger. In our work, the experimentally measured

inner diameters of the CNTs filled with NbSe₂ are 1.9-3.2 nm. These CNTs are all much smaller than 6 nm and do not fall into the range where the flat tubes are the lower energy configuration.

2. For small-diameter CNTs, their flattening only depends on the activation of external factors, such as pressure, temperature, and mechanical strain. Zhang et al. reported the collapse of 2.6 nm single-walled CNTs due to intertube vdW force and thermal activation by rapid cooling from 1023 K (Zhang C, Bets K, Lee SS, et al. ACS Nano, 2012, 24, 6, 6023-32). In our work, after an annealing treatment from a similar temperature of 973 K, all the unfilled CNTs in Fig. 1a remain circular. Especially in Fig. 1a, the diameter of the single-walled empty tube (red box) is larger than that of the neighboring NbSe₂@CNT (yellow box), but no radial deformation occurs. Besides, we have added the STEM characterization of the cross-section of the empty CNTs, which remain circular with an inner diameter of 2.7 nm as shown in Fig. R2. Therefore, we can rule out that the flattening is due to the CNTs themselves, temperature activation and intertube vdW forces. In the absence of other external factors, only one factor, the pressure during the reaction, can break the thermodynamic equilibrium state and transition the CNT to a flattened configuration.

Figure 1a. Atomic-resolved cross-sectional BF images of single-layered flattened NbSe₂ nanotube. Scale bar: 2 nm.

Figure R2. Cross-sectional BF-STEM image of pristine CNTs, Scale bar: 2 nm.

3. Self-contraction of carbon materials filled with other (non-carbon) materials can build up extreme pressure in the interior. Sun et al. reported the collapse of CNTs filled with transition metals such as Fe, Co or Ni deforms the crystals and eventually extrudes them. They claimed a pressure of 20–40 GPa measured from the lattice spacings of the elastically compressed metal cores and estimated from atomistic simulations (Sun L. et al. *Science*, 2006, 312, 1199-1202). Wang et al. observed lattice compressive strain in Mo₂C filled inside CNTs, and attributed it to the strong Mo-CNT interaction and compressive effect by the concave surface of CNTs (Wang K. Xia Y, Jin C, et al. *Journal of the American Chemical Society*, 2023, 145, 23, 12760-12770). Recently, Zeng et al. trapped the volatiles such as argon and neon in nanostructured diamond capsules, and not only retain high-pressure materials, but also the 'pressure' itself (Zeng Z, Wen J, Lou H, et al. *Nature*, 2022, 608, 513-517). These suggest that CNTs are capable of generating and retaining high degrees of internal pressure.
4. The NbSe₂ configuration determines the degree of pressure exerted by CNTs. As shown in Fig R3, we observe an abrupt change in the number of NbSe₂ layers with the varied CNT diameter. Voids can be observed in the region of the diameter mutation (circled in red oval), which are excess spaces pulled out by the unbalanced pressure of the CNT with different degrees of deformation. If only thermodynamics is considered, the atoms are supposed to fill the internal space uniformly. Therefore, we insist that the radial deformation of the CNTs in our system does exert pressure on their interior.

Figure R3. Side-viewd BF-STEM image of NbSe₂@CNTs, Scale bar: 2 nm.

2. Raman spectroscopy on ensemble samples was presented as another indicator for pressure. However, Raman measurement on ensemble samples, which contains various states of sample encapsulation and different diameters and number of walls in carbon nanotube containers, is not convincing. The Raman peak shift from NbSe₂

encapsulation could also originate from different reasons, such as the doping effect. To remove the uncertainty, the authors must also consider the effect of doping and others. Moreover, the authors must present better Raman data with well-defined sample configurations, such as the same-diameter carbon nanotube with/without NbSe₂ encapsulation.

Reply: Thank you for your comments. We agree that further elaborating on the interactions using CNTs of the same diameter would be helpful. However, the preparation of CNTs of the same diameter has been an unsolved challenge for researchers in the field of carbon nanotubes, which limits us from performing subsequent experiments. Then, we tried our best to separate the samples but failed due to the extremely strong vdW forces between the CNTs. Other researchers also found that it is very difficult to obtain an individual CNT from ensembled samples. For this reason, we chose to decouple the Raman peak shift to distinguish its origin and remove the uncertainty.

Raman spectroscopy has been a useful tool in characterizing strain as changes in lattice constants lead to variations in phonon frequencies, but are also affected by doping effects. Despite the fact that the characteristic G and 2D Raman peak shifts are highly sensitive to both strain and doping effects, it is worth noting that their fractional variation ($\Delta\omega_{2D}/\Delta\omega_G$) due to strain (2.2 ± 0.2) is quite different from that induced by doping (0.70 ± 0.05) (Lee J, Ahn G, Shim J, et al. Nature Communications, 2022, 3, 1024). In our work, both G and 2D peaks were significantly shifted compared to the empty CNTs even though the ensemble samples we tested contain CNTs of different chirality, different diameters, and different wall numbers. Under characterization conditions similar to those in the literature, we optically separate mechanical strain from charge doping by exploiting the $\Delta\omega_{2D}/\Delta\omega_G$ difference (Fig. R4). Raman shifts are mainly determined by strain effect, and although a one-to-one correspondence is not possible, it is reasonable to use average shifts to estimate the CNTs strain. We have added “These shifts, decoupled by using fractional variations, are assumed to be dominated by strain rather than doping effects.” in the revised manuscript and Figure S12 in the supplementary information, and we hope it could be acceptable for you.

Figure R4. Decoupling of Raman modes for samples taken at 532 nm laser excitation wavelength.

3. The authors also performed the electrical measurements. Although it seems that the authors tried their best, the electrical transport measurements on the carbon nanotube bundles are not a good way to discuss the intrinsic properties of carbon nanotubes. (1) the nanotube-nanotube junction will significantly influence the electrical transport properties but was not considered at all in the manuscript.

Reply: Thank you for your comments. Electron transport at low temperature in CNTs has been well described by Luttinger Liquid theory, which has experimentally been confirmed in contacts between CNTs (Phys. Rev. B 2000, 62, R10653, Phys. Rev. Lett. 2004, 92, 216804, Phys. Rev. Lett. 2018, 121, 047702), CNT ropes (Science 1997, 275, 1922–1925, Phys. Rev. B 2004, 69, 195406), and even in CNT films (Phys. Rev. B 2001, 64, 233401, Solid State Commun. 2003, 127, 215–218). In this work, due to the difficulty of CNTs separation, we used fiber samples for electrical transport measurements, but we strived to ensure their similar diameters and lengths. The reviewer suggests that the nanotube-nanotube junction influences the electrical transport properties, however, we have shown a comparison of two unfilled CNTs (Fig. 4d). Due to the different electrical properties in terms of conductivity, these two samples should have different distributions and numbers of junctions. Nevertheless, the Luttinger exponent α of two samples are the same, probably due to the dense and close contact between CNTs resembles connecting them in parallel. We therefore conclude that nanotube-nanotube junctions in our samples have no impact on α at low temperatures, and we have added a discussion of Fig. 4d in the manuscript “The power-law plot of unfilled CNTs shows the difference in conductance of different samples, suggesting dissimilarities in the number and distribution of CNT-CNT junctions in the ensembled samples (Fig. 4d). However, a consistent Luttinger exponent α_{CNT} of 0.64 was demonstrated at high temperature (10-100 K), which is in agreement with previous research³⁴, implying a strong response to the intrinsic electron transport properties of CNTs. Remarkably, the electrical curve of

NbSe₂@CNT appears to be steeper approaching the low-temperature limit. At low temperature (2-10 K), the fitting parameter of NbSe₂@CNT gives a much larger exponent of $\alpha_{\text{NbSe}_2@\text{CNT}} \approx 0.32$ than that of unfilled CNTs ($\alpha_{\text{CNT}} \approx 0.22$) and NbSe₃@CNT ($\alpha_{\text{NbSe}_3@\text{CNT}} \approx 0.18$)”.

Figure. 4d Power law relation of conductance G with temperature ($G \sim T^\alpha$) for two unfilled CNT samples from 10-100 K.

(2) In particular, the surface functionalization or doping from the NbSe₂ encapsulation process may cause unexpected behaviors.

Reply: Since the encapsulation process is in a vacuum quartz ampoule, the deselenization reaction is carried out under Ar and H₂ atmospheres, and the heating and cooling process is also continuously protected by Ar, there is not an issue of surface functionalization of CNTs caused by the oxygen-containing atmosphere. In the above discussion on Raman spectroscopy, we also explained that there is almost no doping effect of NbSe₂ on CNTs. We provided different NbSe₂@CNT samples for several electrical transport measurements, and the fitting exponents are all larger than those of unfilled CNTs (Fig. S21). This is a strong indication of the significant enhancement of the electron interactions due to the internal pressure after filling with NbSe₂, while the fluctuation of α possibly arises from the pressure difference induced by the different NbSe₂ configurations.

Figure. S21. Power law relation of conductance G with temperature for three NbSe₂@CNT samples from 2-100 K.

4. Finally, it needs to be clarified the origin of the existence of high pressure from the synthesis process.

(1)The authors explained that the deselenization step induces pressurization, but why?

Reply: Thank you for your question. In this work, we suppose that the pressure during reaction process comes from the inward contraction of the CNTs. According to the statistical results in Fig.S8, the number of Nb atoms in the sample cross-section increased by 50-120% after the deselenization reaction, thus the density of the filler inside the CNTs of the same diameter increased significantly based on stoichiometric ratio. Due to interatomic interactions, the significant increase in the number of atoms inevitably brings about a huge radial volume expansion of the filler, exerting outward expanding pressure on the CNTs. In turn, CNTs contract inwardly and counteracts on the NbSe₂ to maintain a high pressure state. In addition, since NbSe₃ is a one-dimensional chain structure, there is no anisotropic force in its radial direction. While NbSe₂ is intrinsically two-dimensional, applying a radially single-direction force to its externally wrapped CNT. As the compression modulus of NbSe₂ is higher than the elastic modulus of CNTs, the tube appears to be flattened. Therefore, the deformation of the CNT becomes a visualization of the pressure under the dimensional effect.

Figure. S8. The Nb atom numbers of the encapsulated NbSe₂ and NbSe₃ in the cross-sectional STEM images versus the inner diameter of CNTs.

(2) Could the starting material, NbSe₃, be also under pressure?

Reply: We speculate that the starting material NbSe₃ is not under pressure. Fig.S1 showed that all of the carbon tubes filled with NbSe₃ are circular. Pham et al. observed that electromigration/thermal excitation forces drive the NbSe₃ chain slide individually or collectively (Pham T, Zettle A. Phys. Status Solidi B, 2019,256, 1900241). Moreover, there is no radial deformation within a CNT where the empty tube region is of the same diameter as the filled region (Fig. R5), thus we conclude that NbSe₃ is hardly pressurized.

Figure. S1. Representative cross-sectional HAADF-STEM images of the NbSe₃ chains encapsulated within CNTs, showing different close-packing patterns associated with the inner diameters of the CNT host. Scale bars: 5 nm.

Figure R5. Side-viewed ADF-STEM image of NbSe₃@CNTs, Scale bar: 2 nm.

(3) What is the deriving mechanism for such an energetically unfavored process (to a high pressurized state) if not?

Reply: Based on our calculations, the deselenization reaction is not an energetically unfavorable process. At 973 K, the Gibbs free energy difference for the deselenization reaction is less than 0, which suggests a thermodynamically spontaneous process. Subsequently, the volume expansion brought about by the radial rearrangement of Nb and Se atoms and the outstanding compression function of CNTs drive the sample into a high pressurized state. Other researchers also evidenced that high-pressure gases (Ugarte D, Chatelain A, A de Heer W. *Science*, 1996,274: 1897-1899.), Ice VIII and IX occur only at pressures of > 1GPa (Monika J, Monika J, Anatoly I, et al. *Phys Chem Chem Phys*, 2011, 13: 9008 – 9013), and B2-type high-pressure phase KI nanocrystals (Urita K, Shiga Y, Fujimori T, et al. *J Am Chem Soc*, 2011, 133: 10344-10347) can be trapped in carbon nanospaces at ambient pressure. Therefore, we consider this process to be spontaneous under experimental conditions and to result in a high-pressure state.

In summary, we have added a detailed explanation of the pressure origin in the revised manuscript as follows: “Considering that the system is at atmospheric pressure and moderate constant temperature, the calculated Gibbs free energy difference based on the bulk material is negative ($\Delta G < 0$), and thus the deselenization reaction, $NbSe_3 \rightleftharpoons NbSe_2 + Se$, is thermodynamically favorable that drives it proceed spontaneously.” and “Upon elevating the temperature, the enhanced diffusion facilitates the free migration of atoms inside the chemically inert CNTs and enables the in-plane growth of NbSe₂. According to the statistical analysis of the experimental images before and after deselenization, the number of Nb atoms increases significantly (50%-120%) in CNTs with similar equivalent diameters, thus bringing about a significant volume expansion in the radial direction. The intrinsically anisotropic growth behavior of 2D materials will result in expansion along only one direction, thus exerting asymmetric pressure on the CNT inner walls. In turn, the CNT capsule, composed of ultra-strong C-C bonds, spontaneously contracts to counteract

the internal expansion, and thus pressurizes its contents (Fig. S10).”

Referee #3

Thank you for your recommendation of our manuscript. We are grateful to see your positive feedback on the novelty and significance of this work. Your comments and suggestions are really valuable and helpful for revising and improving our manuscript. Based on your comments and suggestions, we made a revision on the manuscript and answered your questions and suggestions as follows. After this revision, we sincerely ask you to consider our revised manuscript for the further publication.

1. My only concern with the manuscript is that no statistical analysis is provided to demonstrate the concentration of either the chains or the resulting flat tubes inside the nanotubes (ie, what percentage of the tubes is filled). This however has an important role in trying to understand whether the measured differences in conductance are indeed associated with filling of the tubes (and hence the NbSe structures) or whether it may arise from other sources related to f.ex. high-temperature annealing. Judging from Fig. 1a it appears that the filling rate is not very high, since this image is presumably from an area displaying the observed structures the best.

Reply: Thank you for your comment. We agree that the filling rate should be mentioned in the manuscript. The filling ratio of NbSe₂ is only 5-10%, however, the starting material NbSe₃ was filled to 85-90%. We speculate that the intrinsic properties (diameter and chirality) of the CNTs determine the encapsulated NbSe₃ close-packed/spiral-packed state, which in turn determines the ability to transform into NbSe₂ flat tubes. Under the thermal excitation and airflow blowing, the NbSe₃ chains that cannot be converted slip away, and those that are converted into NbSe₂ flat tubes are retained due to the huge radial volume expansion plugging the CNTs. Despite a large difference in filling ratios, NbSe₃@CNTs and NbSe₂@CNTs exhibit opposite behavior relative to CNTs. If the confined NbSe₃ slips out completely, the Luttinger coefficient should rise to be consistent with that of the unfilled CNTs. However, the Luttinger coefficient of NbSe₂@CNT is much higher than that of CNTs. We therefore conclude that this experiment is sufficient to illustrate the modulation of electronic interactions under the interior pressure.

According to your suggestion, we have revised the manuscript as follows: “The encapsulated NbSe₃ chains within the CNTs prepared by the CVT method were used as starting materials, and the filling ratio of NbSe₃ chains is 85-90% (Fig. S1-S5).”, “Furthermore, we found that the conversion of the NbSe₃ chains seems to be correlated with the intrinsic chirality of the CNTs, only 5-10% can be converted to the NbSe₂ flat tube.”

2. Spectroscopic characterization of the structures (for example through EELS mapping) would provide additional evidence supporting the interpretation of the atomic structure of the NbSe₂ tubes.

Reply: Thank you for your suggestion. We have added EELS mapping into the supplementary information as shown in Figure R6, which has shown spatial matching with the ADF-STEM image.

Figure R5. ADF-STEM of NbSe₂@CNT and corresponding chemical elemental mapping of C, Nb, and Se by Electron Energy Loss Spectroscopy (EELS). Scale bar: 2 nm.

Reviewers' Comments:

Reviewer #1:

Remarks to the Author:

The authors have addressed my comments. I recommend the manuscript for publication.

Reviewer #2:

Remarks to the Author:

My previous comments and concerns were mostly well addressed by the revision. I support its publication in Nature Communications.

Reviewer #3:

Remarks to the Author:

In my view the manuscript can be accepted for publication.